# Glycemic effect of post-meal walking compared to one prandial insulin injection in type 2 diabetic patients treated with basal insulin: A randomized controlled cross-over study

Onnicha Suntornlohanakul[1¤], Chatvara Areevut[1], Sunee Saetung[1], Atiporn Ingsathit[2,3], Chatchalit Rattarasarn[1]*

1 Division of Endocrinology and Metabolism, Department of Medicine, Ramathibodi Hospital, Mahidol University, Bangkok, Thailand, 2 Division of Nephrology, Department of Medicine, Ramathibodi Hospital, Mahidol University, Bangkok, Thailand, 3 Department of Clinical Epidemiology and Biostatistics, Ramathibodi Hospital, Mahidol University, Bangkok, Thailand

¤ Current address: Division of Endocrinology & Metabolism, Department of Medicine, Prince of Songkla University, Songkhla, Thailand
* chatchalit.rat@mahidol.ac.th

## Abstract

Studies demonstrate that post-meal walking decreases postprandial hyperglycemia in type 2 diabetic patients but it has never been tested with the active treatment comparator. The objective of this study was to determine the effect of post-meal walking on glycemic control compared with one prandial insulin in type 2 diabetic patients who failed basal insulin. A randomized controlled cross-over study of post-meal walking or one prandial insulin was done in type 2 diabetic patients who were being treated with basal insulin between May 2017 and March 2018. In post-meal walking group, patients walked after meal for 15–20 minutes one meal a day every day for 6 weeks. In prandial insulin (basal plus) group, one prandial insulin was injected before breakfast or main meal with rapid-acting insulin. The primary outcome was a difference in HbA1c reduction in post-meal walking compared with basal plus groups. Fourteen patients completed the study. By intention-to-treat analysis, HbA1c was reduced by -0.05(range:-1.08 to 0.74) and -0.19(range:-0.8 to 0.56) % in post-meal walking and basal plus groups respectively. By per-protocol analysis, post-meal walking and basal plus groups decreased HbA1c by 0.13(range:-0.74 to 1.08) and 0.2(range:-0.56 to 0.8) %, respectively. There ~~was~~ were no significant differences in HbA1c reduction from baseline in each group and between groups in both intention-to-treat and per-protocol analysis. Fructosamine levels were decreased by 17.5(-59 to 43) and 10(-15 to 40) μmol/L, respectively at 3 and 6 weeks in post-meal walking group whereas the respective changes in basal plus group were 12.5(-17 to 64) and 17.5(-28 to 38) μmol/L and there were no significant differences in fructosamine reduction from baseline in each group and between groups. In conclusion, although post-meal walking might be as effective as one prandial insulin to improve glycemic control in type 2 diabetic patients who failed basal insulin but the magnitude of

**Data Availability Statement:** All relevant data are within the paper and its Supporting Information files.

**Funding:** This study was supported by research fund to OS by Faculty of Medicine, Ramathibodi hospital, Mahidol university, Bangkok, Thailand. The funder had no role in study design, data collection and analysis, decision to publish, or preparation of the manuscript.

**Competing interests:** The authors have declared that no competing interests exist.

reduction was small. A longer-term study with a larger sample size or with a different walking protocol is required.

## Introduction

Several diabetes guidelines [1, 2] recommend initiation of basal insulin in type 2 diabetic patients after failure to oral hypoglycemic drugs (OHD). If fasting plasma glucose (FPG) is lowered to appropriate ranges but HbA1c has not reached the target, adding medications to reduce the postprandial hyperglycemia is the next step. Adding rapid-acting insulin before one main meal (basal-plus) or GLP-1 agonist in combination with basal insulin or switching from basal to pre-mixed insulin are the options. However, with all of the above options, the patients need more than one injection a day and may expose to increased risk of hypoglycemia. Moreover, GLP-1 agonist is expensive and has irritable gastrointestinal adverse effects.

Increased physical activity is recommended as the mainstay therapy for type 2 diabetic patients especially those who are overweight or obese [3, 4]. Recent studies demonstrate that increased physical activity by walking after meal (post-meal walking) for 10–20 minutes can reduce postprandial plasma glucose (PPG) better than walking before meal [5–12]. Colberg et al [12] showed that post-dinner walking in type 2 diabetic subjects decreased PPG at 1 hour after meal about 40 mg/dl compared with those without. Pahra et al [8] and Reynolds et al [9] respectively demonstrated that HbA1c and glycated albumin were reduced with 10–15 minutes walking after three main meals for two and eight weeks. The effect on PPG reduction has been observed since first time of walking and is not insulin dependent [10]. Nevertheless, none of the previous studies compares PPG-lowering effect of post-meal walking with that of the active comparators.

This study aimed to compare the efficacy of post-meal walking with one prandial insulin on glycemic control in type 2 diabetic patients who failed basal insulin therapy.

## Material and methods

### Study participants

Type 2 diabetic patients aged 35–70 years who were treated with at least one OHD and basal insulin (NPH or Determir or Glargine or Degludec) were recruited from outpatient clinics at Ramathibodi hospital. Patients who had FPG less than 150 mg/dl and HbA1c levels between 7–9% were included. The exclusion criteria were uncontrolled hypertension (systolic blood pressure > 160 or diastolic blood pressure > 100 mmHg), recent myocardial infarction or ischemic stroke within 3 months, chronic lung diseases or heart failure, foot problems (severe diabetic neuropathy, fracture, deformity, previous amputation) which were obstacle to walking, currently took systemic steroids, alcohol consumption more than 7 drinks per week or caffeine consumption more than 400 mg/day, travel regularly across time zone or perform shift work. All participants gave written informed consent. The protocol was approved by the Ethical Clearance Committee, Faculty of Medicine, Ramathibodi hospital, Mahidol university and was registered with Thai Clinical Trials Registry (TCTR20170419003) which is one of the World Health Organization's International Clinical Trials Registry platform. The study conformed to the provisions of the Declaration of Helsinki.

### Study protocol

The study was a randomized controlled cross-over study conducted between May 2017 and March 2018 at outpatient clinic, Ramathibodi hospital, Bangkok, Thailand. The study

comprised of 2 weeks of run-in, 6 weeks of interventions and 2 weeks of wash-out periods prior to cross over (S1 Fig). The study protocol can be followed by the link https://doi.org/10. 17504/protocols.io.72chqaw.

**Run-in period.**  At the 1st visit of run-in period, the participants received and were instructed how to use glucose meter (Freestyle Optium H, Abbot, USA), and accelerometer (Triaxial accelerometer, Fitbit zip, Fitbit, USA). This accelerometer device has been validated for its accuracy in several trials [13–17]. Diabetes education which included insulin injection, detection and correction of hypoglycemia, self-monitoring blood glucose (SMBG), and diet and food choices were provided to all participants. Those food choices included serving and portion size as well as the compositions of nutrients.

During the 1st week of run-in period, the participants recorded the diet in food diary and performed SMBG 6 times for one day (before and 2 hours after breakfast, lunch and dinner). The participants also carried the accelerometer during the day to monitor the steps. In the 2nd week, the participants revisited our clinic to review the food diary and glucose meter use. In this week, participants must walk for 15–20 minutes at least one meal per day. The walk should be started 15–30 minutes after meal. The accelerometer was carried during the day to confirm the walking steps after meal.

**Randomization.**  At the end of the run-in period, participants were randomized into the post-meal walking or the basal plus groups by computerized generated block of 4 randomization using program stata version 15.1. OS enrolled and assigned participants to each intervention.

**The post meal walking group.**  The participants must walk for 15–20 minutes after meal at least one meal per day every day. The speed of walk should be "Walk as fast as possible". During the day, the accelerometer was used to monitor the number of walking steps.

**The basal plus group.**  The participants were advised to use rapid acting insulin analog (Glulisine, Sanofi, France) within 15 minutes before the main meal as prandial or bolus insulin. The starting dose was 4 units/meal or 0.1 unit/kg body weight depending on participants' characteristic. Bolus insulin dose was adjusted as appropriate via mobile phone or line application within 2 weeks until 2-hour PPG $\leq$ 180 mg/dl. In this group, the participants could do their usual activity but they were advised not to walk after meal.

During the study period, if the participants developed hyperglycemic emergency (diabetic ketoacidosis, hyperglycemic hyperosmolar state), had FPG > 250 mg/dl or HbA1c > 9%, the participants would be excluded from the study.

## Measurements

In one day of each week, the participants of both groups recorded the food diary and performed SMBG 6 times a day (before and 2 hours after breakfast, lunch and dinner). The participants continued their usual doses of OHD and basal insulin. If there was evidence of hypoglycemic event, the physicians would adjust the regimen as appropriate.

The participants visited the clinic at 0, 3, and 6 weeks of each intervention to have blood tests, reviewed the food diary, SMBG records, accelerometer use and self-care.

Blood samples were measured in the central laboratory of Ramathibodi hospital in the same day of the tests. Plasma glucose was measured by Hexokinase/Glucose-6-Phosphate Dehydrogenase method (Abbot, USA). HbA1c was measured by turbidimetric inhibition immunoassay (Roche, Germany). Fructosamine was measured by chemiluminescence method (Roche, Germany).

## Outcomes

The primary outcome was a difference in HbA1c reduction from baseline between basal plus and post-meal walking groups. The hypothesis was that HbA1c reduction between two groups was not different. The main secondary outcome was a difference of fructosamine reduction from baseline between groups.

## Statistical analysis

The sample size was calculated using the formula:

$$n = \frac{\left(Z_{1-\frac{\alpha}{2}} + Z_{1-\beta}\right)^2}{\Delta^2}\sigma^2.$$

The number of participants required to detect a 0.5% HbA1c difference was calculated. A paired-tests with an alternative hypothesis mean of 0.5 and a standard deviation of 0.65 requires a sample size of 14 to attain 80% power, assuming a two-sided alpha of 0.05. Estimation of HbA1c variation with 0.65% standard deviation was used according to previous study from Lankisch et al [18]. Assuming there would be 15% dropout rate, we designed to include 16 participants into the study. The analysis was based on statistical concept of the cross-over study. Multilevel mixed-effects linear regression was used for analysis of the normal distribution outcomes, a fixed effect model with treatment, sequence and period entered into the model and subjects was a random effect. The median regression analysis was used for the non-normal distribution outcomes. All data was analyzed by the STATA program version 15.

## Results

Nineteen participants were recruited between May 2017 and March 2018 but five withdrew from the study during run-in period (S1 Fig). The baseline characteristics of 14 participants were shown in Tables 1 and S1. About two-third of participants used insulin glargine as a basal insulin. Sulfonylurea (85%), metformin (71%) and dipeptidyl peptidase 4 inhibitors (50%) were the main OHD in this study. Pioglitazone, sodium-glucose co-transporter 2 inhibitors and alpha-glucosidase inhibitors were used in four (28.6%), two (14.3%) and one (7.1%) participants, respectively.

**Table 1. Baseline characteristics of participants.**

| Baseline characteristics (N = 14)[a] | |
|---|---|
| Female | 8 (57.1%) |
| Age (years) | 56.93 ± 1.82 |
| Weight (kg) | 77.29 ± 5.48 |
| BMI (kg/m$^2$) | 29.45 ± 1.66 |
| Waist circumference (cm) | 99.43 ± 3.92 |
| Duration of diabetes (years) | 9 (1–22) |
| HbA1c (%) | 7.90 ± 0.12 |
| Fasting plasma glucose (mg/dl) | 131.29 ± 6.94 |
| Dose of basal insulin (units) | 16 (6–44) |
| Numbers of oral hypoglycemic drugs | 2.5 ± 0.23 |

[a] Data are expressed as mean ± SE or median (range)

The participants in post-meal walking group had significantly greater total daily steps, total post-meal walking steps and time spending in post-meal walking after lunch and dinner than those in basal plus group. Most of participants in post-meal walking group walked after dinner. None of participants in basal plus group walked after meals. Concerning bolus insulin, eight participants used bolus insulin at breakfast, five at dinner and one at lunch. Both total carbohydrates and caloric intake per day of post-meal walking group were significantly greater than those of basal plus group particularly at lunch time (S2 Table).

As shown in Fig 1, by intention-to-treat analysis, HbA1c was reduced by -0.05 (range: -1.08 to 0.74) and -0.19 (range:-0.8 to 0.56) % in post-meal walking and basal plus groups respectively. However, by per-protocol analysis, the respective reduction in HbA1c was -0.13 (range: -0.74 to 1.08) and -0.20 (range: -0.56 to 0.8) %. There were no significant differences of HbA1c reduction from baseline in each group and between groups in both intention-to-treat and per-protocol analysis. In post-meal walking group, fructosamine levels was decreased from baseline at 3 and 6 weeks whereas in basal plus group, it was decreased only at 3 week. However, the magnitude of the overall fructosamine reduction from baseline was not different in each group and between groups. (Fig 2)

The glucose excursion of SMBG at 3 and 6 weeks were decreased from baseline in both groups but did not reach statistical significance (S2 and S3 Figs). Few episodes of documented hypoglycemic events (glucose <70 mg/dl) occurred in two participants. There was no other adverse events in each group.

## Discussion

Our results indicated that in patients with type 2 diabetes who were being treated with basal insulin, the HbA1c reduction by post-meal walking or one prandial insulin injection were not different at 6 weeks. Shorter term glycemic control as indicated by fructosamine levels was improved by post-meal walking group but was not significantly different from baseline and was not different from that of basal plus group. Therefore, post-meal walking may be as effective as one prandial insulin in control of PPG in type 2 diabetic patients who fail from basal insulin at least in short-term basis. To our knowledge, this was the first study that evaluated the role of post-meal walking on PPG control compared with standard prandial insulin therapy in free-living diabetic patients.

However, it should be noted that HbA1c reduction at 6 weeks of post-meal walking group was quite small. There are several plausible reasons to explain this unexpected finding. First, 15 min of walking after one meal in this study may not be long enough for meaningful reduction of PPG. The study by Van Dijk et al [11] demonstrated that walking after meal for 15 min although could reduce PPG in type 2 diabetic patients, but the reduction did not reach statistical significance. In other studies, at least 20 minutes of walking after one meal or 10–15 minutes per each meal everyday demonstrated significant glycemic reduction [8, 9, 12, 19]. Pahra et.al [8] demonstrated that post-meal walking for 1,500–1,600 steps for 15 min after three main meals could significantly decrease the HbA1c levels of 0.9% at 8[th] weeks. In our study, the post-meal walking steps after dinner which was the main meal of walking was only 700 steps. These indicate that the amount of steps and times spending in post-meal walking of our study may be inadequate. Second, patients assigned for post-meal walking in our study may not comply with the walking protocol. It could be noted from S2 and S3 Tables that some patients did not walk after meal. As seen from Fig 1, five of 14 participants had increasing HbA1c levels after 6-week of post-meal walking. Of these five, three did not comply with walking protocol. Therefore, the efficacy of post-meal walking in terms of HbA1c reduction may be underestimated. If we used per-protocol analyses and excluded these three participants, there

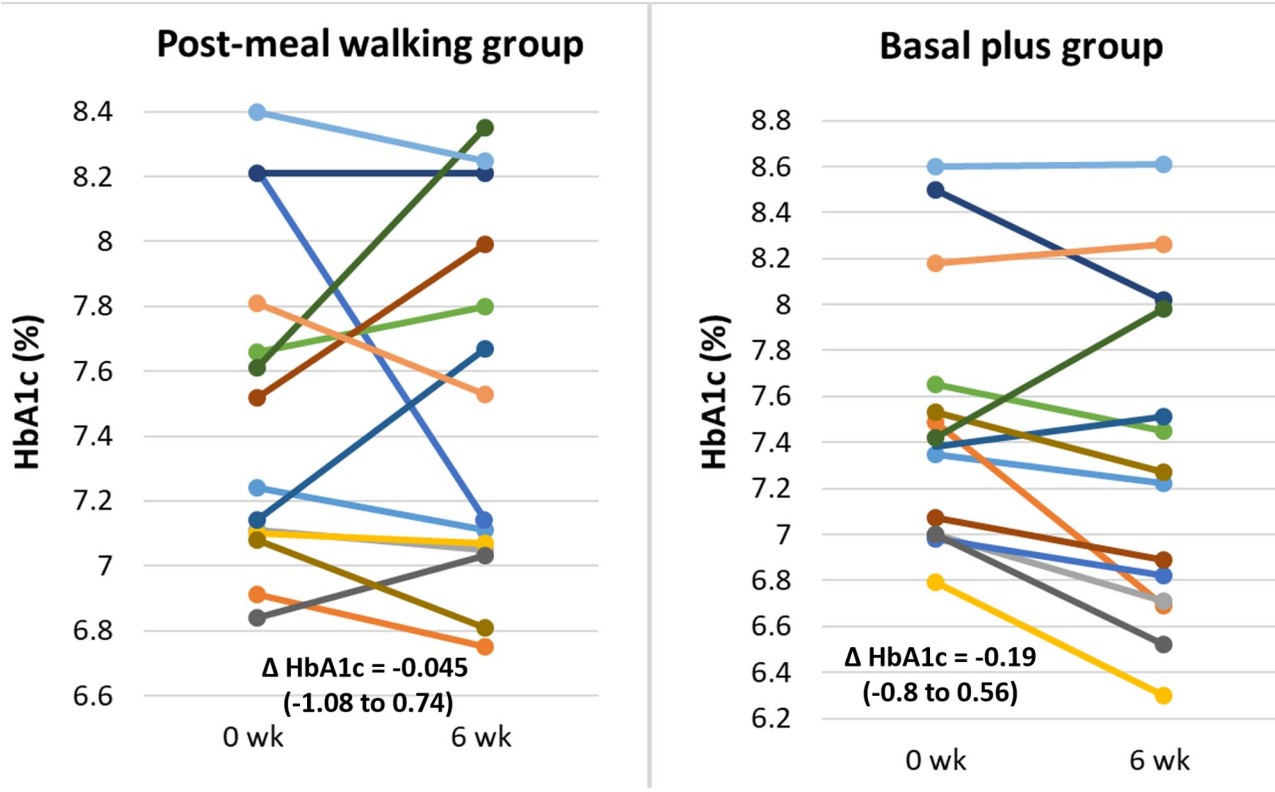

**Fig 1. Changes of HbA1c levels of the post-meal walking and basal plus groups by intention-to-treat analysis.** Δ HbA1c is expressed as median (range).

would be 0.13 and 0.20% reduction of HbA1c in post-meal walking and basal plus group, respectively. It could have been possible that if our subjects were strict to walking protocol, the magnitude of HbA1c reduction in post-meal walking group would be greater. Third, time and the amount of meal at which walking is performed is also an important issue. Several studies performed walking after breakfast [5–7], of which maximum glucose excursion occurred [20]. A study by Reynold et al [9] showed that walking after meal with the most substantial amount of carbohydrate, could decrease PPG more effectively than walking after other meals. Since most participants walked after dinner and dinner was not a main meal in our study (S2 Table), the magnitude of PPG reduction with post-meal walking may be substantially under-estimated. Fourth, given the daily total carbohydrate and caloric intake of post-meal walking group were significantly greater than those of basal plus group, the effect of walking on HbA1c reduction may also be under-estimated. It should be noted that the magnitude of HbA1c reduction was also modest in basal plus group as compared with other previous studies [18, 21]. This may be due to a shorter time of treatment compared with other studies.

Our study had several limitations. First, we used 6 points blood glucose monitoring data from once weekly SMBG which might not be adequate to represent a real, daily blood glucose fluctuation. Second, the duration of the study was not long enough to capture the exact changes of HbA1c levels. However, we tried to solve this problem by using fructosamine levels. Furthermore, the study results may be influenced by carry over effects due to the short intervention and wash-out periods chosen. However, there was no difference of HbA1c at the end (6th week) of first intervention and at the start (0 week) of the second intervention groups (HbA1c 7.51 *vs* 7.53%, P = 0.878) in participants who started with post-meal walking and in

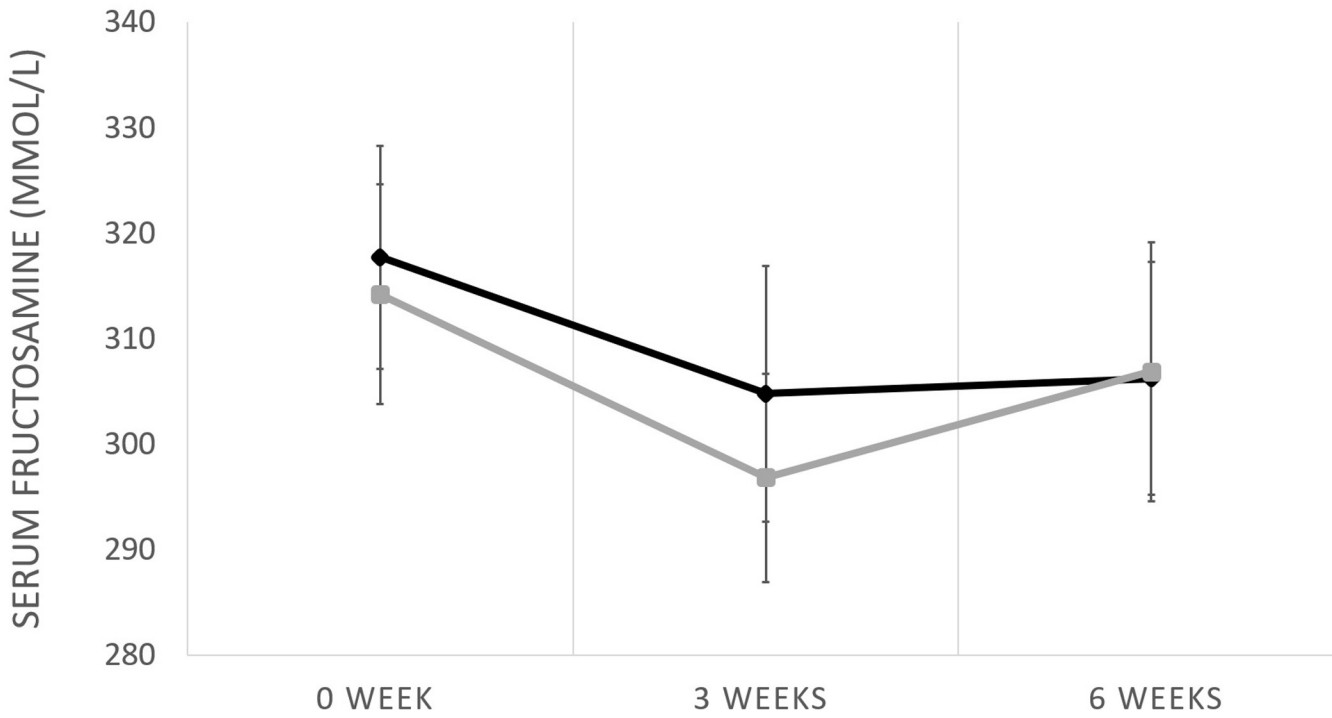

**Fig 2. Changes of serum fructosamine levels in post-meal walking and basal plus groups.** Data are expressed as mean ± SE.

those who started with basal plus (HbA1c 7.27 *vs* 7.30%, P = 0.885). Third, the sample size in this study was small. A significant number of our participants did not comply with the walking protocol. The strengths of this study were that it was the first and the longest study that compared post-meal walking with prandial insulin in basal insulin treated type 2 diabetic patients with valid study design. Most previous studies were performed in the controlled, experimental conditions and had no active treatment as a comparator.

## Conclusion

This study indicated that post-meal walking of at least one meal per day might be as efficient as one mealtime insulin to improve glycemic control in type 2 diabetic patients who failed from basal insulin therapy. Nevertheless the magnitude of glycemic reduction appears to be small. A longer-term study with a larger sample size and/or with a different walking protocol is required to test the efficiency and effectiveness of post-meal walking in glycemic control of type 2 diabetic patients in real world.

## Supporting information

**S1 Table. Baseline characteristics of each participant in the study.**
(DOCX)

**S2 Table. Dietary intake, walking steps and time spending in walking in post-meal walking and basal plus groups.**
(DOCX)

**S3 Table. Insulin dosages, post-meal walking steps and duration and plasma glucose levels changes by prandial insulin or post-meal walking of each participant in the study.**
(DOCX)

**S1 Fig. The study protocol.**
(TIF)

**S2 Fig. Self-monitoring blood glucose results of post-meal walking group.**
(TIF)

**S3 Fig. Self-monitoring blood glucose results of basal plus group.**
(TIF)

**S1 Checklist. CONSORT 2010 checklist.**
(PDF)

**S1 File. Study protocol-Thai version.**
(PDF)

**S2 File. Study protocol-English translation.**
(PDF)

## Acknowledgments

The contribution in the statistical analysis by Ms.Sukanya Siriyotha from Department of Clinical Epidemiology and Biostatistics, Ramathibodi hospital, Mahidol university and Assistant Professor Prapasri Kulalert, MD from Faculty of Medicine, Thammasart university were acknowledged.

## Author Contributions

**Conceptualization:** Chatchalit Rattarasarn.

**Data curation:** Onnicha Suntornlohanakul.

**Formal analysis:** Atiporn Ingsathit.

**Funding acquisition:** Chatchalit Rattarasarn.

**Investigation:** Onnicha Suntornlohanakul, Chatvara Areevut, Sunee Saetung.

**Methodology:** Atiporn Ingsathit.

**Project administration:** Onnicha Suntornlohanakul.

**Supervision:** Chatchalit Rattarasarn.

**Writing – original draft:** Onnicha Suntornlohanakul.

**Writing – review & editing:** Chatchalit Rattarasarn.

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
