## [Decision Letter · Decision Letter 0]

20 Sep 2019

PONE-D-19-20612

Glycemic effect of post-meal walking compared to one prandial insulin injection in type 2 diabetic patients treated with basal insulin: a randomized controlled cross-over study”

PLOS ONE

Dear Professor Rattarasarn,

Thank you for submitting your manuscript to PLOS ONE. After careful consideration, we feel that it has merit but does not fully meet PLOS ONE’s publication criteria as it currently stands. Therefore, we invite you to submit a revised version of the manuscript that addresses the points raised during the review process.

We would appreciate receiving your revised manuscript by Nov 04 2019 11:59PM. To enhance the reproducibility of your results, we recommend that if applicable you deposit your laboratory protocols in protocols.io, where a protocol can be assigned its own identifier (DOI) such that it can be cited independently in the future. For instructions see: http://journals.plos.org/plosone/s/submission-guidelines#loc-laboratory-protocols

We look forward to receiving your revised manuscript.

Kind regards,

Raffaella Buzzetti, M.D.

Academic Editor

PLOS ONE

Journal Requirements:

2. Please include a copy of Table 2 which you refer to in your text on page 11.

Reviewers' comments:

Reviewer's Responses to Questions

**Comments to the Author**

1. Is the manuscript technically sound, and do the data support the conclusions?

Reviewer #1: Yes

Reviewer #2: Partly

2. Has the statistical analysis been performed appropriately and rigorously? 

Reviewer #1: Yes

Reviewer #2: No

3. Have the authors made all data underlying the findings in their manuscript fully available?

Reviewer #1: Yes

Reviewer #2: No

4. Is the manuscript presented in an intelligible fashion and written in standard English?

Reviewer #1: Yes

Reviewer #2: No

5. Review Comments to the Author

Reviewer #1: This is an interesting study comparing the effects of prandial insulin with post-prandial walking. Anything that encourages exercise ahead of insulin should be encouraged.

I think that we should regard this as a pilot study which requires further work and hopefully publication will stimulate this.

However:

• The study was only over a very short period. HbA1c will not have had time to change as significantly as it could have over say 12 weeks.

• This was a group of patients were on a variety of OHDs – I realise that they were controls against themselves but I am surprised that some patients remained on sulphonylureas. I assume that the basal insulin (type and dosage) remained the same for both legs.

• Only 14 out of 19 completed the study. Within these groups some in the walking post-meal group did not walk and is there any guarantee that in the group that had the prandial insulin that they did not walk post meal?

• 700 steps is not very far and I suspect rather than aiming for “Walk as fast as possible” setting a target of possibly 1000 or 1500 steps would have been better.

• I am not sure why the time that patients were asked to walk was not after the same meal that they were going to take their prandial insulin – surely that would have been a better comparison?

• I may have missed this - did any of the patients develop DKA or HHS?

• Only fructosamine and HbA1c were measured. Lipids, BP, weight etc would have been useful. Particularly a Q of L questionnaire would have been interesting to see if patients preferred the walking or the prandial insulin.

• It would have better if the groups could have been controlled for calorific intake. Eating more at lunch in the walking group will have diluted the results.

Reviewer #2: The authors adress whether post-meal walking may be as effective to lower HbA1c as once daily prandial insulin administerd before the main/largest meal in patients with type 2 DM and inadequate glycemic control (A1c > 7.0%) despite treatment with oral hypoglycemic drugs and basal insulin. A cross-over design with two six weeks treatment periods and a 2 weeks washout period is empoyed. A mixed model is employed to test for significance between the two treatment types/periods.

Major concerns:

Although the authors adress an important and relevant question, the study design is flawed by the duration of the study periods and the primary endpoint (A1c) chosen. The authors acknowledge that their design with 6 weeks treatment periods and a 2 weeks washout-period carries a significant risk of carry-over effects. The discussed fact, that the A1c did not change during the washout period it is not helpful at all since even significant changes in glycemic excursions are unlikely to change A1c concentrations within this short time period. Since the authors seem to be aware of this, either a different primary endpoint or a different study design should have been chosen.

Minor concerns:

Statistical analysis: the details of the the mixed model (i.e. parameters chosen for fixed, random effects, period, etc.) should be indicated in order to make the analysis plan reproducible.

The patients were instructed to use prandial insulin before the main meal (basal plus period) or to briskly walk after at least one meal (post-meal walking period). To judge the comparative effects of the 2 interventions compareable time periods should be analyzed. I.e., if prandial insulin was given before dinner, this period should also be looked at in the post meal walking group and data should be presented accordingly.

Detailed data regarding the insulin therapy should be indicated (i.e. dose, time-course of dose escalation) should be indicated in order to judge whether appropriate dose adjustments were made.

The caloric intake in the basal plus group was significantly lower. Can the authors offer an explanation for this finding ?

Some english language editing is adviseable.

The individual participant data used for the final analysis should be made available.

6. PLOS authors have the option to publish the peer review history of their article (what does this mean?). If published, this will include your full peer review and any attached files.

Reviewer #1: Yes: Prof Andrew Collier

Reviewer #2: No

---

## [Author Response · Author response to Decision Letter 0]

8 Oct 2019

Responses to reviewer #1

1. The study was only over a very short period. HbA1c will not have had time to change as significantly as it could have over say 12 weeks.

Answer: It is well known that HbA1c is an estimate of mean plasma glucose levels in the previous 3-4 months, however 50% of the result is from mean plasma glucose level 0-30 days or 1 month prior to blood sampling (Can J Diabetes 2013; 37: S35-S39). Therefore we believe 6-week of intervention should have significant change (if any) to HbA1c results. We’re afraid that if the intervention of interest (post-meal walking) is continued for 12 weeks to capture all changes of HbA1c, the participants might not cooperate and we might have lost some of the participating subjects during the trial.

2. This was a group of patients were on a variety of OHDs – I realise that they were controls against themselves but I am surprised that some patients remained on sulphonylureas. I assume that the basal insulin (type and dosage) remained the same for both legs.

Answer: It is a common practice in Thailand that sulphonylureas is not withdrawn while being on basal insulin. As shown in the S3 table, basal insulin type was not changed. Basal insulin dosage was stable in all except two subjects in each arm, the dosage was minimally reduced due to mild hypoglycemia.

3. Only 14 out of 19 completed the study. Within these groups some in the walking post-meal group did not walk and is there any guarantee that in the group that had the prandial insulin that they did not walk post meal?

Answer: Since all participants carried accelerometers during the trial, we could monitor whether they walked after prandial insulin injection or not. It was shown that none of the participants in the basal plus arm violated the protocol. We have added this statement in the Results. 

4. 700 steps is not very far and I suspect rather than aiming for “Walk as fast as possible” setting a target of possibly 1000 or 1500 steps would have been better.

Answer: We absolutely agree with you. The greater the number of post-meal steps, the lower of post-meal glucose levels is anticipated but in real life, walking for 1,000-1,500 steps post-meal might not be practical for some patients particularly with elderly patients. We try to make our protocol feasible and practical. However, we re-analyzed individual data of each participant and found that those who complied with the walking protocol, the averaged post-meal walking steps was in fact more than 1,000 steps as shown in S3 Table.

5. I am not sure why the time that patients were asked to walk was not after the same meal that they were going to take their prandial insulin – surely that would have been a better comparison?

Answer: We absolutely agree with this comment but in real life, the free time that patients have to spend on walking does not always get along with the main meal. As having been said, we try to make it practical. 

6. I may have missed this - did any of the patients develop DKA or HHS?

Answer: None of the participants developed hyperglycemic emergencies during the trial 

7. Only fructosamine and HbA1c were measured. Lipids, BP, weight etc would have been useful. Particularly a Q of L questionnaire would have been interesting to see if patients preferred the walking or the prandial insulin.

Answer: Unfortunately we did not collect those data 

8. It would have better if the groups could have been controlled for calorific intake. Eating more at lunch in the walking group will have diluted the results.

Answer: We agree but unlike in clinical research environment, controlling for caloric intake in free living condition is difficult and make the study protocol more complicated

Responses to reviewer#2

Major concerns:

Although the authors address an important and relevant question, the study design is flawed by the duration of the study periods and the primary endpoint (A1c) chosen. The authors acknowledge that their design with 6 weeks treatment periods and a 2 weeks washout-period carries a significant risk of carry-over effects. The discussed fact, that the A1c did not change during the washout period it is not helpful at all since even significant changes in glycemic excursions are unlikely to change A1c concentrations within this short time period. Since the authors seem to be aware of this, either a different primary endpoint or a different study design should have been chosen.

Answer: As similar to our responses to reviewer#1, we believe 6-week of intervention should have significant change to HbA1c results since half of A1c change would occur in the first 30 days prior to blood sampling although it cannot capture all changes (Can J Diabetes 2013; 37: S35-S39). Fructosamine level which can reflect the shorter term changes of mean plasma glucose levels may be a better choice in this regard but since its measurement method is not well standardized and it is not a standard measure of glucose control in clinical practice, therefore we decide to use fructosamine as a secondary endpoint of the study.

Minor concern

1. Statistical analysis: the details of the the mixed model (i.e. parameters chosen for fixed, random effects, period, etc.) should be indicated in order to make the analysis plan reproducible.

Answer: The details of the mixed model statistical analysis were added in the manuscript as suggested

2. The patients were instructed to use prandial insulin before the main meal (basal plus period) or to briskly walk after at least one meal (post-meal walking period). To judge the comparative effects of the 2 interventions comparable time periods should be analyzed. I.e., if prandial insulin was given before dinner, this period should also be looked at in the post meal walking group and data should be presented accordingly.

Answer: As you can see in the S3 Table, the meals at which the patients walked or injected prandial insulin were not always the same, therefore it is impossible to analyze such data. We designed our protocol to be practical in real life, so we did not fix the meal at which post-meal walking was performed.

3. Detailed data regarding the insulin therapy should be indicated (i.e. dose, time-course of dose escalation) should be indicated in order to judge whether appropriate dose adjustments were made.

Answer: Insulin dose and changes of plasma glucose levels post-meal by walking or prandial insulin were demonstrated in S3 Table

4. The caloric intake in the basal plus group was significantly lower. Can the authors offer an explanation for this finding?

Answer: This finding may occur by chance since we did not control the caloric intake of the participants during the trial. 

5. The individual participant data used for the final analysis should be made available.

Answer: The individual participant data was shown in S3 Table

Other inquiries

Please include a copy of Table 2 which you refer to in your text on page 11.

Answer: It is a mistake, Table 2 is actually a S2 Table. This has been corrected in the revised manuscript.

 Please note that we have corrected some mistake in the abstract (line 33 page 2).

 The study protocol has been deposited to protocol io and its DOI has been added in the manuscript as recommended.

---

## [Decision Letter · Decision Letter 1]

28 Jan 2020

PONE-D-19-20612R1

Glycemic effect of post-meal walking compared to one prandial insulin injection in type 2 diabetic patients treated with basal insulin: a randomized controlled cross-over study”

PLOS ONE

Dear Professor Rattarasarn,

Thank you for submitting your manuscript to PLOS ONE. After careful consideration, we feel that it has merit but does not fully meet PLOS ONE’s publication criteria as it currently stands. Therefore, we invite you to submit a revised version of the manuscript that addresses the points raised during the review process.

Please, address the remaining minor issues raised by the reviewers.

We would appreciate receiving your revised manuscript by Mar 13 2020 11:59PM. To enhance the reproducibility of your results, we recommend that if applicable you deposit your laboratory protocols in protocols.io, where a protocol can be assigned its own identifier (DOI) such that it can be cited independently in the future. For instructions see: http://journals.plos.org/plosone/s/submission-guidelines#loc-laboratory-protocols

We look forward to receiving your revised manuscript.

Kind regards,

Noël C. Barengo, MD, PhD, MPH

Academic Editor

PLOS ONE

Reviewers' comments:

Reviewer's Responses to Questions

**Comments to the Author**

1. If the authors have adequately addressed your comments raised in a previous round of review and you feel that this manuscript is now acceptable for publication, you may indicate that here to bypass the “Comments to the Author” section, enter your conflict of interest statement in the “Confidential to Editor” section, and submit your "Accept" recommendation.

Reviewer #1: All comments have been addressed

Reviewer #2: All comments have been addressed

Reviewer #3: (No Response)

2. Is the manuscript technically sound, and do the data support the conclusions?

Reviewer #1: Yes

Reviewer #2: Yes

Reviewer #3: Yes

3. Has the statistical analysis been performed appropriately and rigorously? 

Reviewer #1: Yes

Reviewer #2: Yes

Reviewer #3: Yes

4. Have the authors made all data underlying the findings in their manuscript fully available?

Reviewer #1: Yes

Reviewer #2: Yes

Reviewer #3: Yes

5. Is the manuscript presented in an intelligible fashion and written in standard English?

Reviewer #1: Yes

Reviewer #2: Yes

Reviewer #3: Yes

6. Review Comments to the Author

Reviewer #1: The concept of exercise in place of insulin is an interesting and useful one.

This should be regarded as a pilot and encourage others to undertake further work.

There are faults with the design in this study but cannot be changed now.

The authors answer all the points made by the reviewers.

Reviewer #2: Although the authors acknowledge the major concerns of the reviewers in their response letter no appropriate changes have been made to the manuscript. The authors should clearly state in their conclusion that the study results may be influenced by carry over effects due to the short inntervention and wash-out periods Chosen.

Reviewer #3: A randomized controlled cross-over study was conducted in patients (n=14) with type 2 diabetes with the goal of determining the effect of post-meal walking on glycemic levels compared to prandial insulin. The change in HbA1c was the primary outcome. There were no significant differences in HbA1c from baseline to follow-up in with or between the groups.

Minor revisions:

1- Abstract: Please clarify the following statement since both confidence intervals contain zero.It seems reasonable that neither group showed a statistically significantly decrease over baseline.

“By per-protocol analysis, post-meal walking and basal-plus groups significantly decreased HbA1c by 0.13(range:-0.74 to 1.08) and 0.26(range:-0.8 to 0.08) %, respectively.”

2- Line 145: Indicate if the alpha level was one- or two-sided. The beta for a power of 0.80 is 0.20. Please clarify.

3- Line 146: This statement is technically incorrect. “At least 14 participants were needed to demonstrate the statistical significance.” Possibly something to this effect would be more appropriate, “A paired-tests with an alternative hypothesis mean of 0.5 and a standard deviation of 0.65 requires a sample size of 14 to attain 80% power, assuming a two-sided alpha of 0.05.” Possibly this test should be one-sided since a decrease in HbA1c is expected.

4- Line 149: Indicate the underlying covariance structure used in the linear mixed models and the criteria for choosing it.

5- Line 189-191: Provide the overall p-value for comparing times 0, 3 and 6. If the overall p-value is significant use a multiple comparison tests to summarize pairwise differences.

6- Both Figures 1 and 2 display nearly identical data. Include only one of these.

7. PLOS authors have the option to publish the peer review history of their article (what does this mean?). If published, this will include your full peer review and any attached files.

Reviewer #1: No

Reviewer #2: Yes: Stefan Bilz

Reviewer #3: No

---

## [Author Response · Author response to Decision Letter 1]

12 Feb 2020

Responses to reviewer #1. The concept of exercise in place of insulin is an interesting and useful one. This should be regarded as a pilot and encourage others to undertake further work. There are faults with the design in this study but cannot be changed now.

The authors answer all the points made by the reviewers.

Answer: No specific query raised by the reviewer.

Responses to reviewer#2. Although the authors acknowledge the major concerns of the reviewers in their response letter no appropriate changes have been made to the manuscript. The authors should clearly state in their conclusion that the study results may be influenced by carry over effects due to the short intervention and wash-out periods chosen.

Answer: We have addressed the concerning issue raised by reviewer #2 in the Discussion part.

Responses to reviewer#3.

1. Abstract: Please clarify the following statement since both confidence intervals contain zero. It seems reasonable that neither group showed a statistically significantly decrease over baseline.

“By per-protocol analysis, post-meal walking and basal-plus groups significantly decreased HbA1c by 0.13(range:-0.74 to 1.08) and 0.26(range:-0.8 to 0.08) %, respectively.”

Answer: This statement has actually been corrected in the first revised manuscript but may have been missed. The statement that has already been in the Abstract is “There was no significant differences in HbA1c reduction from baseline in each group and between groups in both intention-to-treat and per-protocol analysis” 

2. Line 145: Indicate if the alpha level was one- or two-sided. The beta for a power of 0.80 is 0.20. Please clarify.

3. Line 146: This statement is technically incorrect. “At least 14 participants were needed to demonstrate the statistical significance.” Possibly something to this effect would be more appropriate, “A paired-tests with an alternative hypothesis mean of 0.5 and a standard deviation of 0.65 requires a sample size of 14 to attain 80% power, assuming a two-sided alpha of 0.05.” Possibly this test should be one-sided since a decrease in HbA1c is expected.

Answer: Those statements have been corrected and revised according to the reviewer’s suggestion.

4. Line 149: Indicate the underlying covariance structure used in the linear mixed models and the criteria for choosing it.

Answer: We explore data in each patient and it showed random intercept and random slope. So we decided to choose random coefficient model for our outcome analysis.

5. Line 189-191: Provide the overall p-value for comparing times 0, 3 and 6. If the overall p-value is significant use a multiple comparison tests to summarize pairwise differences.

Answer: We thank the reviewer for this suggestion. Actually there was no overall change of fructosamine levels after intervention, therefore the result of fructosamine measurements has been revised in the Result and Discussion sections. Figure of fructosamine results has also been revised accordingly.

6. Both Figures 1 and 2 display nearly identical data. Include only one of these.

Answer: Fig 2 has been removed and replaced with Fig 3.

---

## [Decision Letter · Decision Letter 2]

4 Mar 2020

Glycemic effect of post-meal walking compared to one prandial insulin injection in type 2 diabetic patients treated with basal insulin: a randomized controlled cross-over study”

PONE-D-19-20612R2

Dear Dr. Rattarasarn,

We are pleased to inform you that your manuscript has been judged scientifically suitable for publication and will be formally accepted for publication once it complies with all outstanding technical requirements.

With kind regards,

Noël C. Barengo, MD, PhD, MPH

Academic Editor

PLOS ONE

Additional Editor Comments (optional):

Reviewers' comments:

Reviewer's Responses to Questions

**Comments to the Author**

1. If the authors have adequately addressed your comments raised in a previous round of review and you feel that this manuscript is now acceptable for publication, you may indicate that here to bypass the “Comments to the Author” section, enter your conflict of interest statement in the “Confidential to Editor” section, and submit your "Accept" recommendation.

Reviewer #2: All comments have been addressed

Reviewer #3: All comments have been addressed

2. Is the manuscript technically sound, and do the data support the conclusions?

Reviewer #2: Yes

Reviewer #3: (No Response)

3. Has the statistical analysis been performed appropriately and rigorously? 

Reviewer #2: Yes

Reviewer #3: (No Response)

4. Have the authors made all data underlying the findings in their manuscript fully available?

Reviewer #2: Yes

Reviewer #3: (No Response)

5. Is the manuscript presented in an intelligible fashion and written in standard English?

Reviewer #2: Yes

Reviewer #3: (No Response)

6. Review Comments to the Author

Reviewer #2: (No Response)

Reviewer #3: (No Response)

7. PLOS authors have the option to publish the peer review history of their article (what does this mean?). If published, this will include your full peer review and any attached files.

Reviewer #2: Yes: Stefan Bilz

Reviewer #3: No

---

## [Editor Report · Acceptance letter]

11 Mar 2020

PONE-D-19-20612R2 

Glycemic effect of post-meal walking compared to one prandial insulin injection in type 2 diabetic patients treated with basal insulin: a randomized controlled cross-over study” 

Dear Dr. Rattarasarn:

I am pleased to inform you that your manuscript has been deemed suitable for publication in PLOS ONE. Congratulations! Your manuscript is now with our production department. 

With kind regards,

on behalf of

Dr. Noël C. Barengo 

Academic Editor

PLOS ONE